# Outbreak of *Pityogenes chalcographus* and *Pityophthorus pityographus* on Spruce Seedlings Resulting from Inappropriate Management in a Forest Nursery

**Tomáš Fiala** * and **Jaroslav Holuša**

Faculty of Forestry and Wood Sciences, Czech University of Life Sciences, Kamýcká 129,
16501 Praha, Czech Republic; holusa@fld.czu.cz
* Correspondence: tomas.fiala@nature.cz; Tel.: +420-724-151-113

**Abstract:** In this report, we describe a local outbreak of small bark beetles on 4-year-old seedlings of *Picea abies* in a forest nursery in Central Europe in 2021. In March 2022, 10–50% of the seedlings were "dry" (i.e., with dry and easily broken twigs and with easily peeled bark) in each 4-row bed in the nursery. Half of the dry plants were completely covered by galleries of bark beetles and another 35% of the seedlings were with one or two bark beetle galleries. Almost 90% of the beetles found on the affected seedlings were *Pityogenes chalcographus*, and only 10% were *Pityophthorus pityographus* (we studied 100 seedlings in the second bed). The chipping of *Picea pungens* in previous years had left many felled trunks, branches, and other logging residues in the area. These residues are covered by galleries of both bark beetles. We suggest that, after multiplying on the logging residues, the beetles could not find suitable material for reproduction and were lured to the nursery seedlings, which had been weakened by location (a location that provided inadequate shade and no wind protection) and by repeated replanting.

**Keywords:** bark beetle; chipping; logging residues; *Picea abies*; *Picea pungens*; Scolytinae




## 1. Introduction

Bark beetles (Coleoptera: Curculionidae: Scolytinae) are currently the most important forest pests in both Eurasia and North America. The main bark beetle pests are species of *Ips* in Eurasia and species of *Dendroctonus* in North America [1]. These species infest host trees, attract other beetles, and create dense systems of galleries in stems in which females lay eggs. Hatched larvae consume the phloem of mature trees, which results in tree mortality. Bark beetles in these genera are mostly monophagous, but some species are oligophagous or polyphagous [2]. Economically less important and smaller species compared to those mentioned in the previous sentences occur on the thinner parts of trees. These small species can be abundant, but they do not kill adult trees [2].

Many authors have recorded seedling mortality resulting from feeding by bark beetles. Only bark beetles in the genus *Hylastes* are recognized as serious pests of coniferous seedlings [2,3]. Most of the reported bark beetle infestations of coniferous seedlings have involved the widely distributed genus *Pityophthorus* (Eichhoff, 1864 [1]). The reasons for the increase of *Pityoptorus* bark beetles abundance and subsequent infestation of seedlings are not mentioned (Table A1).

This genus has a total of 20 species and subspecies in Europe, including two invasive species, *P. juglandis* Blackman, 1928 and *P. solus* Blackman, 1928 [4]. Species of the genus *Pityophthorus* are secondary pests that invade weakened trees or withering parts of trees [5]. Only a few reports have documented *Pityophthorus* bark beetles as primary pests of mature trees: *P. confertus* Swaine, 1917 and *P. confinis* LeConte, 1876 caused the deaths of several dozen *Pinus ponderosa* in the U.S.A. [6]; infestation of healthy Douglas-fir (*Pseudotsuga menziesii*) twigs by *P. orarius* Bright, 1968 reduced tree fertility [7]; and *P. carmeli* Swaine,

1918 and *P. setosus* Blackman, 1927 can damage healthy twigs of *Pinus radiata* D. Don and transmit the fungus *Fusarium circinatum* Nirenberg & O'Donnell, which causes Pitch Canker Disease [8].

In Europe, the most common species in this genus is *P. pityographus* Ratzeburg, 1837, which occurs in high abundance on all species of European conifers, mostly on thin branches <2 cm in diameter [2,9,10]. Economic damage caused by this species has been previously reported only once [11]. More damage in southern Europe is caused by *P. ramulorum* Perris, 1856 (syn. *pubescens* Marsham, 1802), which carries the fungus *F. circinatum* [12]. Along with a fungus that it transmits *P. juglandis* causes "thousand cankers disease" in Italy [13].

Data regarding the occurrence of *Pityophthorus* bark beetles on seedlings in Europe are limited to two very early papers: Nüsslin [14] reported an infestation of pine seedlings by *P. lichtensteinii* Ratzeburg, 1837, and Escherich [15] reported infestations on spruce seedlings by *P. pityographus* and *P. exculptus* Ratzeburg, 1837.

In this report, we describe the local outbreak of two species of small bark beetles, *P. pityographus* and *Pityogenes chalcographus* (Linnaeus, 1761), in a forest nursery (Figure 1). We also consider the possible causes of the outbreak.

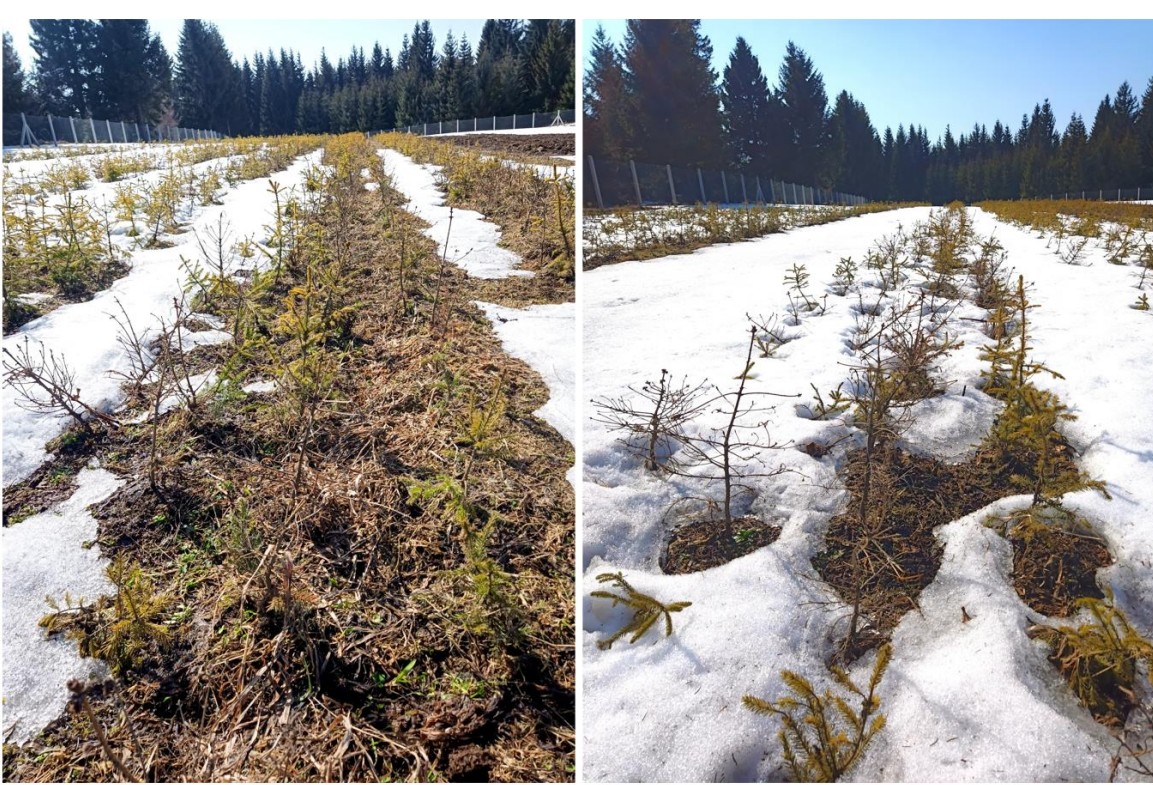

**Figure 1.** "Dry" seedlings in a forest nursery near the village of Kovářská in March 2022.

## 2. Materials and Methods

### 2.1. Locality

The forest nursery at Kovářská (Central Europe: 50.4290047 N, 13.0515328 E) is located in the forest of the Krušné Hory Mts. at 860 m above sea level (asl). The nursery occupies 1.5 ha, and has 10 beds, each with four rows of 4-year-old *Picea abies* seedlings. The seedlings were transplanted into the rows when they were 1 year old. The nursery sits on a <5° slope with western exposure (Figure 1). Sun exposure throughout the year lasts from 10:00 to sunset. Westerly winds predominate and affect the nursery because it is not protected by forests to the west. The soil is acidic with signs of glazing. Spruce seedlings were removed from the soil in April 2021, but because the customer did pick them up, the nursery owner reforested them after 2 weeks [16].

### 2.2. Samplings

On 25 March 2022, we evaluated the health of the seedlings in 5 of the 10 beds; these beds were located 1, 3, 5, 8, and 11 m from the western edge of the nursery and, unlike the other 5 beds, these beds were not covered by snow (Figure 1). In each bed, >250 seedlings were rated as "dry" (i.e., with dry breaking twigs and easily peeled bark), "withering" (i.e., with at least one branched with dry, brown, and shriveled needles), or "healthy" (i.e., all twigs with green needles).

In addition, we removed all of the above-ground parts of 100 seedlings in the middle of the second bed, e.g., 3 m from the edge. In the laboratory, the bark and bast of the seedlings were carefully removed with a scalpel, and the identities of all beetles found were determined by T. Fiala by Pfeffer's key [2].

As galleries of bark beetles cannot be distinguished under bark of seedlings, we quantified the infestation of cut seedlings by bark beetles according to the following categories from the most to the least severe: the whole surface was covered by bark beetle galleries; two bark beetle gallery systems were evident; one bark beetle gallery system was evident; the bark beetle nuptial gallery (or galleries) was flooded with resin; and the seedling lacked any evidence of bark beetle infestation (Figure 2).

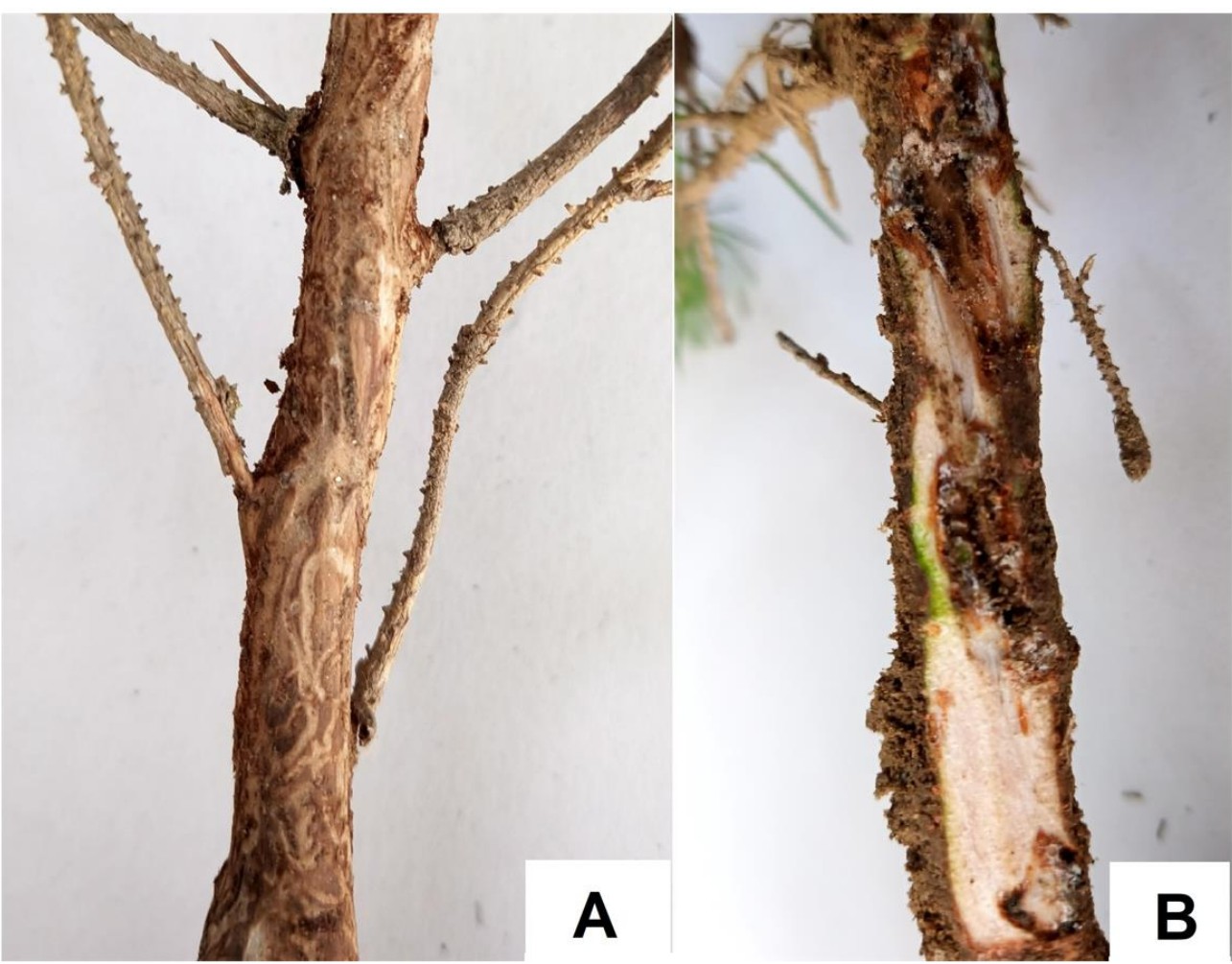

**Figure 2.** Seedling covered by galleries of bark beetles (**A**) and bark beetle nuptial chambers flooded with resin (**B**).

*2.3. Statistical Analyses*

The relationship between the number of dry seedlings and the distance from the western edge of the nursery was analyzed by regression. The thicknesses of the trunks of dry vs. healthy seedlings were compared with a *t*-test in in Statistica 12.0.

## 3. Results

In the five beds that were not covered with snow, a significant percentage of the seedlings were dry. Almost 50% of the seedlings were dry in the first two beds from the western edge of the nursery, and the percentage of dry seedlings in beds decreased with distance from the western edge of the nursery (y = 48.3 − 28.3 × log10(x); r = −0.87; *p* = 0.05). The percentage of withering seedlings decreased (y = 11.6 − 7.5 × log10(x); r = −0.73; *p* = 0.16; Figure 3). In each bed, the percentage of healthy plants increased with distance from the western edge of the nursery (y = 40.1 + 35.8 × log10(x); r = 0.85; *p* = 0.07; Figure 3). However, the relationships are not significant.

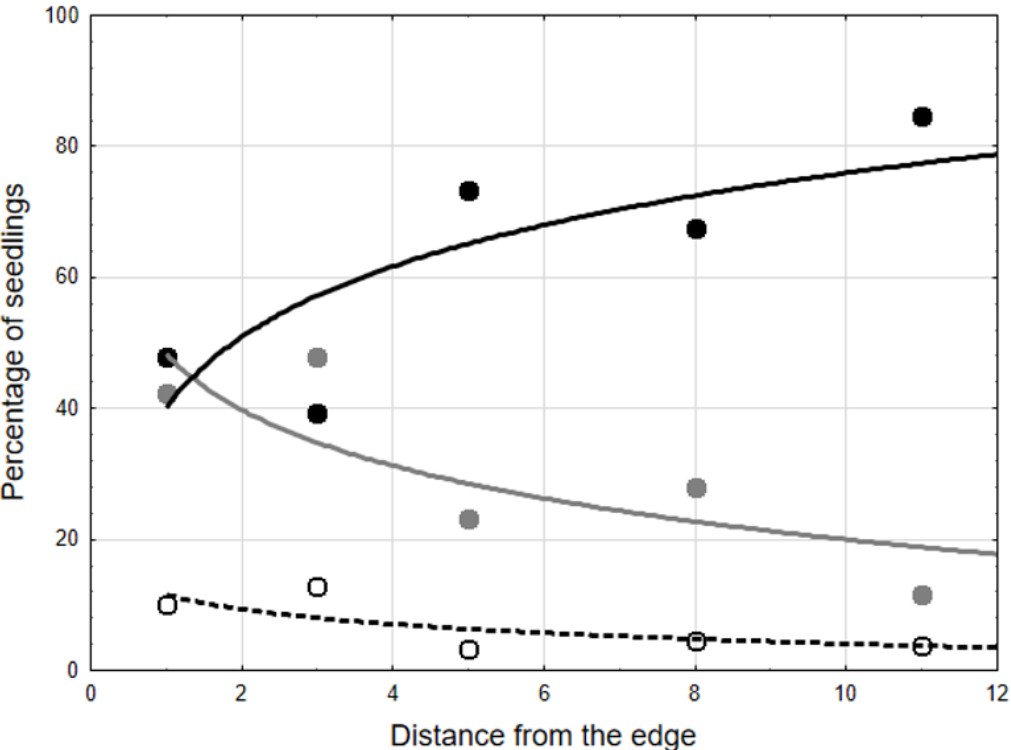

**Figure 3.** Percentage of dry (grey), withering (white), and healthy plants (black) in beds as a function of distance from the western edge of the nursery.

Of the 100 seedlings collected, 56% were dry, 6% were withering, and 38% were healthy. The thickness of stems at the soil surface did not significantly differ between dry and healthy seedlings (t = −0.27; p = 0.79; Table 1). Half of the dry seedlings were completely covered (from the base of the stem-to-stem diameter ≤3 mm) with bark beetle galleries (nuptial chambers, maternal galleries, larval galleries, and pupal chambers) that were deeply cut into wood. Fewer than 20% of the 100 plants had only one or two gallery systems, and almost 10% showed no signs of infestation. Only bark beetle entry holes were found in the remainder of the seedlings.

**Table 1.** Properties of the 100 seedlings that were cut and examined in the laboratory.

| Category of Seedlings | Dry | Withering | Healthy |
|---|---|---|---|
| Numbers of seedlings | 56 | 6 | 38 |
| Stem width (mm) | 5.4 ± 1.5 | 6.3 ± 2.1 | 5.4 ± 1.7 |
| Numbers of plants with *Pityogenes chalcographus* beetles | 40 | 1 | 4 |
| Numbers of plants with *Pityophthorus pityographus* beetles | 9 | 1 | 0 |
| Percentage of dry, withering, or healthy plants with | | | |
| The whole surface covered by galleries | 50 | 0 | 0 |
| Two gallery systems of bark beetles | 16 | 17 | 0 |
| One gallery system of bark beetles | 19 | 33 | 5 |
| Enter holes encapsulated in resin | 6 | 33 | 29 |
| No sign of infestation | 9 | 17 | 66 |

Most healthy, almost 20% of withering, and less than 10% dry plants had no signs of infestation. All results are presented in Table 1.

For the 70 seedlings with signs of infestation (whether dry, withering, or healthy), no beetles were found in 20, and 175 beetles were found in the other 50. One to 10 beetles of *P. chalcographus* were found in >70% of the dry seedlings. Beetles in resin-flooded nuptial chambers were found in one withering and four healthy seedlings (Figure 2). A total of 20 *P. pityographus* beetles, one to five per plant, were found in nine dry plants. *P. pityographus* was found with *P. chalcographus* in five plants and was found alone in four plants.

## 4. Discussion

Only a small percentage of dry *P. abies* seedlings in the nursery lacked signs of bark beetle infestation, and 85% of the dry seedlings (Table 1) had at least one or two bark beetle's gallery systems. In half of the dry plants, all of the phloem had been consumed by bark beetle larvae, indicating that bark beetles were the main cause of mortality for those seedlings.

Infestation of relatively large seedlings by *P. pityographus* can be distinguished from infestation by *P. chalcographus* based on the nuptial chamber, i.e., the nuptial chamber of *P. pityographus*, but not of *P. chalcographus* is visible in the wood of the trunk or a branch of sufficient size [2]. However, *P. pityographus* nuptial chambers were not visible on small seedlings (like those in the current study), because the chambers were cut into the wood due to the thin bast. Almost 90% of the beetles found were *P. chalcographus*, and only 10% were *P. pityographus*, which is consistent with previous reports that *P. chalcographus* is a more aggressive colonizer of trees than *P. pityographus* [17,18].

Among the 100 seedlings that were examined in the laboratory, we found mature or callow beetles, because both species mainly overwinter as adults. In Central Europe, both species begin flight activity in May [2] and can therefore infest replanted seedlings early in May. F1 beetles *P. pityographus* do not emerge until autumn, and this species probably has only one generation per year, while *P. chalcographus* has two to three generations per year [2,19]. In mountain ranges above 800 m, however, bark beetles usually have 1.0 to 1.5 generations per year [20–22], and we speculate that both species had one generation in the studied locality (860 m asl) and that part of the F1 generation overwintered in the galleries.

*Pityogenes chalcographus* is not known as a nursery pest (Table A1), and the current report is the first to document its infestation of nursery trees. *P. chalcographus*, however, is recognized as a serious primary pest of young conifer trees [2,15]. *Pityogenes saalasi* Eggers, 1914 and *Pityogenes. bidentatus* Herbst, 1783 may also be primary pests of young conifers in Siberia and Central Europe [23,24]. Another species in the genus *Pityogenes*, *P. calcaratus* Eichhoff, 1878, has also been reported to infest 3- to 8-year-old *Pinus halapensis* trees [25].

*Pityophthorus pityographus* and *P. chalcographus* are among those bark beetles that primarily attack branches [26–28]. Therefore, the abundances of *P. pityographus* and *P. chalcogra-*

*phus* may increase in *Ips typographus* outbreak areas (Linnaeus, 1758) [29], because *P. pityographus* and *P. chalcographus* multiply on branches and logging residues after *P. abies* trees are salvage logged in response to the outbreaks [28,30,31].

In the current study area, however, *I. typographus* was not very abundant, and there was no salvage logging [16]. However, there is or has been a high proportion of *Picea pungens* in the surrounding forest stands, which have been planted in the area beginning in the 1970s after logging of forest stands that were killed by pollutants [32]. *Picea pungens* has been gradually eliminated; the trees are being chipped immediately after being felled or sometime later (Figure 4). After chipping in stands, branches 1 cm to 5 cm thick and 30 cm to 100 cm long remain on the soil surface. As reported by several other authors c.f. [33–36] and based on our own observations, both *P. chalcographus* and *P. pityographus* multiply on felled trees and logging residues (Figure 4).

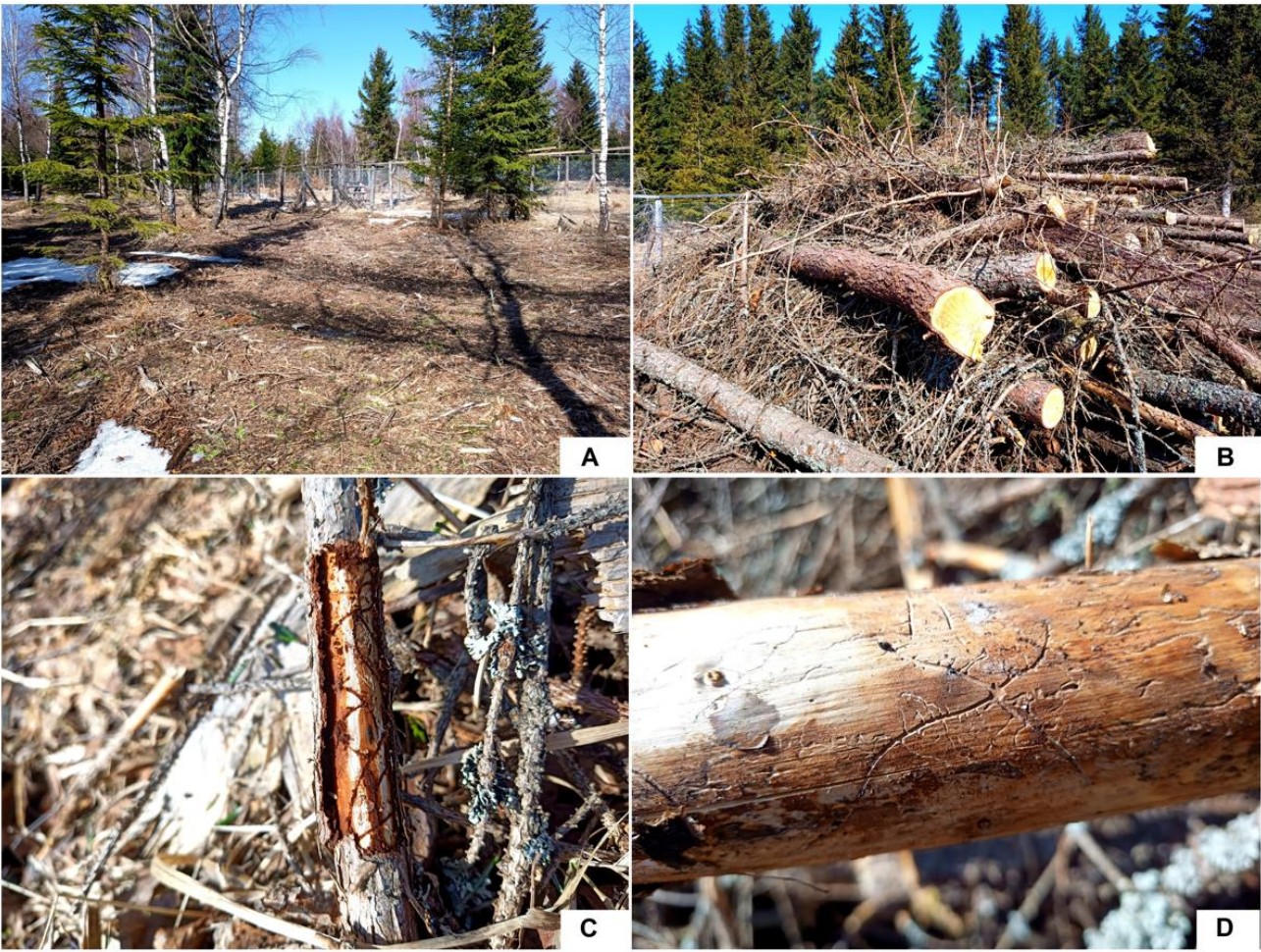

**Figure 4.** Forest after chipping of standing *Picea pungens* trees (**A**) (50.4259353 N, 13.0619114 E); felled *P. pungens* trees at forest stands (**B**) (50.4470644 N, 13.0744211 E); and gallery systems of the bark beetle *P. pityographus* on logging residues (**C**) and on stems of felled *P. pungens* (**D**) in the surroundings of the forest nursery at Kovářská.

We speculate that a large number of newly emerged beetles flew from the residues and searched for suitable breeding sites in 2021 and infested the seedlings in the nursery because better alternatives were not available. Seedlings are more susceptible to stressors than mature trees [37]. In the nursery of the current study, the seedlings were planted at the edge of the forest, which is sunlit all day and unprotected from the wind. These factors certainly dried the soil and consequently weakened the seedlings (Figure 3), which were also stressed by being removed from the nursery soil and then replanted in that

soil. As documented in this report, these seedlings were attacked by *P. chalcographus* and *P. pityographus*, while attack by bark beetles was not observed in the neighboring two nurseries (50.4295 N, 13.0548 E; 50.4291 N, 13.0552 E; personal observations). The latter two nurseries are surrounded by forests and are therefore more shaded and less wind-blown than the current nursery.

## 5. Conclusions

In a forest nursery in the mountains of Central Europe, some of the seedlings were killed by *P. chalcographus* and *P. pityographus*. These two bark beetles became secondary pests because of a large increase in their abundance, but not because of climate change or the salvage logging of mature stands infested by *I. typographus*. The presence of chipping residues and long-lying felled *P. pungens* apparently led to an increase in the abundance of both species. The resulting adult beetles of *P. chalcographus* and *P. pityographus* could evidently not find suitable material for reproduction in the area, and were therefore attracted to the nursery seedlings, which were weakened by the drying of the habitat and by their repeated removal and return to the soil. If the seedlings were not stressed, we doubt that they would have been infested by *P. chalcographus* or *P. pityographus*.

It follows that the infestation of *P. abies* by *P. chalcographus* and *P. pityographus* was probably exceptional rather than typical. Solving this problem will not require the capturing of beetles using aggregation pheromones in traps [38,39] or the treating of seedlings with contact insecticides. Solving this problem will instead require reducing the stress experienced by the seedlings.

**Author Contributions:** The contributions of authors T.F. and J.H. are equal. All authors have read and agreed to the published version of the manuscript.

**Funding:** This research was funded by the Ministry of Agriculture of the Czech Republic, grant number QK1920433.

**Institutional Review Board Statement:** Not applicable.

**Informed Consent Statement:** Not applicable.

**Data Availability Statement:** All data are presented in article.

**Acknowledgments:** The authors thank Bruce Jaffee (USA) for editorial and linguistic improvement of the manuscript, and Milan Hrachovina and Antonín Holeček (LS Klášterec) for cooperation in helping to determine why the seedlings were infested with bark beetles.

**Conflicts of Interest:** The authors declare no conflict of interest.

## Appendix A

**Table A1.** Reports of infestations of coniferous seedlings by bark beetles. Country abbreviations: **AU**—Australia, **CA**—Canada, **CL**—Chile, **CZ**—Czechia, **DE**—Germany, **FI**—Finland, **IL**—Israel, **MX**—Mexico, **NZ**—New Zealand, **PG**—Papua New Guinea, **RO**—Romania, **SE**—Sweden, **UA**—Ukraine, **US**—United States of America, **ZA**—South Africa.

| Bark Beetle Species | Countries | Tree Species | References |
|---|---|---|---|
| *Carphoborus pinicolens* **Wood, 1954** | MX, US | *Abies, Pinus* | [40] |
| *Carphoborus sansoni* **Swaine, 1924** | CA, US | *Picea* | [40] |
| *Cryphalus asperatus* **Gyllenhal, 1813** | CZ | *Picea abies* | [41] |
| *Dendroctonus rhizophagus* **Thomas and Bright, 1970** | MX | *Pinus* | [40] |
| *Hylastes angustatus* **Herbst, 1793** | UA, ZA | *Pinus radiata, P. sylvestris, P. patula, P. elliottii* | [42–45] |

**Table A1.** *Cont.*

| Bark Beetle Species | Countries | Tree Species | References |
|---|---|---|---|
| *Hylastes ater* **Paykull, 1800** | AU, CL, NZ, RO, UA | *Pinus radiata, P. sylvestris, Picea abies* | [45–49] |
| *Hylastes brunneus* **Erichson, 1836** | FI, SE | *Picea abies* | [50,51] |
| *Hylastes cunicularius* **Erichson, 1836** | DE, SE | *Picea abies* | [52,53] |
| *Hylastes linearis* **Erichson, 1836** | IL | *Pinus halepensis* | [54] |
| *Hylastes nigrinus* **Mannerheim, 1852** | US | *Pseudotsuga menziesii* | [55] |
| *Hylastes opacus* **Erichson, 1836** | UA | *P. sylvestris* | [45] |
| *Hylastes salebrosus* **Eichhoff, 1868** | US | *Pinus* | [40] |
| *Hylurdrectonus araucariae* **Schedl, 1964** | PG | *Araucaria cunnighamii* | [56] |
| *Hylurgus ligniperda* **Fabricius, 1787** | AU, CL, UA | *P. radiata, P. sylvestris* | [45,57,58] |
| *Ips paraconfusus* **Lanier, 1970** | US | *P. radiata* | [59] |
| *Pityogenes calcaratus* **Eichhoff, 1878** | IL | *P. halapensis* | [25] |
| *Pityophthorus absonus* **Blackman, 1928** | CA, US | *Abies, Pinus* | [40] |
| *Pityophthorus confertus* **Swaine, 1917** | CA, MX, US | *Pinus* | [40] |
| *Pityophthorus dentifrons* **Blackman, 1922** | CA, US | *Picea* | [40] |
| *Pityophthorus exculptus* **Ratzeburg, 1837** | DE | *Picea* | [15] |
| *Pityophthorus grandis* **Blackman, 1928** | CA, US | *Pinus ponderosa* | [40] |
| *Pityophthorus impexus* **Bright, 1978** | MX | *Pinus* | [40] |
| *Pityophthorus lichtensteinii* **Ratzeburg, 1837** | DE | *Pinus* | [14] |
| *Pityophthorus pityographus* **Ratzeburg, 1837** | DE | *Picea* | [15] |
| *Pityophthorus pseudotsugae* **Swaine, 1918** | CA, US | *Abies, Picea, Pinus, Pseudotsuga, Tsuga* | [40] |
| *Pityophthorus pulchellus tuberculatus* **Bright, 1981** | CA, MX, US | *Picea, Pinus* | [40] |
| *Pityophthorus ramulorum* **Perris, 1856** | IL | *P. halepensis* | [54] |
| *Scolytus monticolae* **Swaine, 1917** | CA, US | *Pseudotsuga menziesii* | [40] |

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
