# Peer review of "Outbreak of Pityogenes chalcographus and Pityophthorus pityographus on Spruce Seedlings Resulting from Inappropriate Management in a Forest Nursery"

_forests, doi:10.3390/f13070987_

Round 1

Reviewer 1 Report

Although it is not a very extensive study, the manuscript »Outbreak of Pityogenes chalcographus and Pityophthorus pityographus on spruce seedlings resulting from inappropriate management in a forest nursery« describes and analyzes an interesting forest health related event.

P. chalcographus is a serious pest of spruce trees, but it is not known as a nursery pest. Although similar events as described in the manuscript have probably occurred several times in the past, there are no official records. An appropriate analysis and description of the injuries mentioned in the manuscript is therefore in place.

However, I have some minor things:

-        Species P. chalcographus was described by Linnaeus in 1761 as Dermestes chalcographus. It was later renamed to Pityogenes chalcographus. Therefore, name of the author and the year next to the name P. chalcographus should be written in brackets = P. chalcographus (Linnaeus, 1761). Check all mentioned bark beetle species and correct if needed!   

-        When a particular species is first mentioned in an article, it is written with its full name, and later the genus name may be abbreviated. Correct throughout the manuscript.

-       Line 70: »the« is written twice

     Line 110: I suggest that exact percentages should be written (instead of >50%, write 56%)

-       Line 132: Latin name of the species should be written in italic

Reviewer 2 Report

The authors report a small study on how improper management may provoke attacks on young seedling trees by small scolytids. It is generally well written, while several plots are problematic in form.

Fig 2

is difficult to see ‘corridors’ (better ‘galleries’) in A or nuptial chambers “w resin” (B). I looked at the photos on-screen at 600% in PDF and by a good color printer. It appears there are problems with resolution (image size), maybe contrast, and some jpeg-losses. I tried to enhance a bit of contrast and color saturation, but it helped little.
Possibly, a better clarity could be found by using only cropped parts, such as a smaller part of bole (larger magnification) of the 2 photos?

Fig 3

is the only data plot, so very important! Data therein is as well analysed partly by regression. For the latter, the x-axis scale shown is not proper (x values not equally spaced nor starting a zero) and the graph as a whole is shown as a ‘business graphs style’ bar graph.
Besides, the regression equation appears incorrect. If you insert x= 10 (m) into the eq.: you get
y= 163 - 20.3x = 163 -203= 
– 40
i.e. a negative value of – 40 (%), while in the plot, all values must be at least 0 (%) to be biologically realistic. Thus, not only is an improper x-axis displayed, but regression numerics are somehow faulty.
Alternatively, despite the 0-value missing one may approximately see when estimating the intercept value (distance 0 m) into eq.:
y= 163 - 0x  = 163 (%),
again a value outside (above) the biologically realistic (>> 100% damage).

I enclose a more proper plot from Excel. Due to the imperfections of Exec, this plot is far from a perfect scientific graph but shows the data for all 3 categories easily comparable and on reasonable scales. I did it by a quick manual reading from an analogue print for data points, so some smaller errors may for sure exist, but plotted points seem to agree well with the values of the bar graph. Note the plot uses the “%” form for a proportion, while the regression lines use the numerical form for a proportion.
Note as well that one of the three series is redundant, as the total sum is 100%, so only 2 are needed to plot.

Fig 4

refers to the pinned specimens of the 2 spp. The value of this fig is unclear, especially as neither the scale of the image nor the sexes of the specimens are given. Why is there a rounded blackish semi-circle in front of (below) the head of the Pp specimen? Pp seems to be described in 1837 – is there a problem with species identification of adults?

Table A1 Line 234:
Has no heading to identify it -is it an Appendix? More commonly Appendix comes after references.

Reviewer 3 Report

The topic is interesting but already quite well known, however it can be re-emphasized if the article has a thorough review in all chapters. The comments are described throughout the pdf

Round 2

Reviewer 2 Report

Revisions are overall well done.

A single exception is L85-88. The sentence is much rewritten and now starts with a phrase that is clearly incorrect as worded (in general we do know how to distinguish bark beetles!). Remedy: 1) Split the sentence into at least 2 parts. 2) First part need several words added to qualify the statement only to the present material and to galleries.

Reviewer 3 Report

Just three small suggestions that should be considered, but nothing important enough not to admit the article for publication
